# Identification of Abnormal Data for Synchronous Monitoring of Transformer DC Bias Based on Multiple Criteria

**DOI:** 10.3390/s23104959

**Published:** 2023-05-22

**Authors:** Zhongqing Kou, Sheng Lin, Aimin Wang, Yuanda He, Long Chen

**Affiliations:** 1School of Electrical Engineering, Southwest Jiaotong University, Chengdu 611756, China; 2School of Electrical Engineering and Electronic Information, Xihua University, Chengdu 610039, China; 3Shenzhen Power Supply Bureau Co., Ltd., Shenzhen 518028, China

**Keywords:** abnormal data identification, transformer DC bias, synchronous monitoring, multiple criteria

## Abstract

Seriously abnormal data exist in the synchronous monitoring data of transformer DC bias, which causes serious data feature contamination and even affects the identification of transformer DC bias. For this reason, this paper aims to ensure the reliability and validity of synchronous monitoring data. This paper proposes an identification of abnormal data for the synchronous monitoring of transformer DC bias based on multiple criteria. By analyzing the abnormal data of different types, the characteristics of abnormal data are obtained. Based on this, the abnormal data identification indexes are introduced, including gradient, sliding kurtosis and Pearson correlation coefficient. Firstly, the Pauta criterion is used to determine the threshold of the gradient index. Then, gradient is used to identify the suspected abnormal data. Finally, the sliding kurtosis and Pearson correlation coefficient are used to identify the abnormal data. Data for synchronous monitoring of transformer DC bias in a certain power grid are used to verify the proposed method. The results show that the accuracy of the proposed method in identifying mutated abnormal data and zero-value abnormal data is claimed to be 100%. Compared with traditional abnormal data identification methods, the accuracy of the proposed method is significantly improved.

## 1. Introduction

With the increasing scale of the metro, a large amount of stray current invades the grid transformer, leading to serious problems of the transformer DC bias [1,2,3]. The transformer DC bias can lead to supersaturation of the transformer core, intensified magnetostriction, increased vibration, abnormal noise and other problems [4,5]. Therefore, transformer DC bias can be understood in time by monitoring the neutral DC, vibration and noise of the transformer [6,7]. However, in the process of synchronous monitoring of transformer DC bias, the synchronous monitoring system is prone to communication failure, electromagnetic interference and other environmental effects [8,9,10], which can lead to abnormal data for the synchronous monitoring of transformer DC bias. Abnormal data in synchronous monitoring of transformer DC bias adversely affect the analysis of transformer DC bias. Therefore, it is of great significance to identify the abnormal data for synchronous monitoring of the transformer DC bias and find the abnormal data in time to solve the problem of transformer DC bias.

At present, researchers have carried out a lot of work in the field of abnormal data identification and have made great achievements. Qiao J et al. proposed a data-driven outlier elimination method by analyzing the characteristics of abnormal data and then combining the “quartile method” and density-based clustering method [11]. Villanueva D et al. proposed a real power curve model by fitting the wind power within each wind speed range to a normal probability distribution, and the data beyond the standard deviation range of three times were identified as outliers [12]. Wang Y et al. assumed that wind power data obey normal distribution, and proposed an abnormal data identification method based on the Pauta criterion [13]. Zheng L et al. combined weighted distance with the local outlier factor algorithm to calculate the outlier factor of a single object, and then used the outlier factor to identify abnormal data [14]. Yang, Z et al., basing their work on the analysis of the weighting relationships among the data, proposed a method to identify the bad data, with a small deviation [15]. Zhao J et al. proposed a robust generalized estimator to identify bad data by exploiting the temporal correlation and the statistical consistency of measurements [16]. Using only the data from the neighboring buses and a one-hop communication system, M. S. Uddin et al. proposed an online probability density-based technique to identify bad data detection in the power grid [17]. Zang Haixiang et al. proposed a bad data identification method based on WGAN-GP (Wasserstein Generative Adversarial Network with Gradient Penalty) [18]. Yan Yingjie et al. proposed an anomaly detection method based on large data analysis such as time series analysis and unsupervised learning, which realized anomaly detection from a new perspective of the data evolution process and data association [19]. Dong Ze et al. proposed an outlier detection method for thermal process by combining a signal decomposition method with a density-based detection method [20]. Li Xinpeng et al. put forward an algorithm based on the isolation forest data anomaly detection method of electric power dispatching flow [21]. The types and variation characteristics of the abnormal data characteristics for the synchronous monitoring of DC bias are greatly different from the above monitoring data. Therefore, it is necessary to propose an identification method that focuses on abnormal data for synchronous monitoring of transformer DC bias.

In view of the above situation, based on the feature of the abnormal data characteristics for synchronous monitoring of DC bias being correlated with the data for synchronous monitoring, the identification of abnormal data for synchronous monitoring of transformer DC bias based on multiple criteria is proposed. The proposed method is suitable for a synchronous and real-time transformer DC bias monitoring system. Firstly, the method collects the normal historical data for synchronous monitoring of transformer DC bias, including neutral DC, vibration and noise. Then, the Pauta criterion is used to calculate the gradient threshold of the above monitoring data, and the gradient is used as a criterion to preliminarily identify abnormal data, normal data and suspected abnormal data. Then, the change of sliding kurtosis and the Pearson correlation coefficient of monitoring data are used as auxiliary criteria to identify suspected abnormal data. Finally, the application analysis shows that this method can accurately and reliably identify the abnormal data for synchronous monitoring of transformer DC bias.

## 2. Analysis of Abnormal Data for Synchronous Monitoring of Transformer DC Bias

The synchronous monitoring system of the transformer DC bias can monitor neutral DC and vibration and noise in real time and synchronously. Figure 1 shows neutral DC monitoring data in a specific power grid.

When the metro is in operation, the neutral DC is larger and has obvious DC bias characteristics, due to the high level of stray currents generated by the metro; similarly, when the metro is in shutdown, the metro does not generate almost any stray currents. The neutral DC amplitude is lower, and the transformer is not under DC bias. In addition, some substations put in a DC blocking device to suppress the transformer DC bias caused by stray currents, which can cause the neutral DC to fluctuate around zero, and the transformer is not under DC bias.

In this paper, based on the synchronous monitoring system of transformer DC bias in a specific power grid, the abnormal data segments from various sets of monitoring information are statistically obtained. By analyzing the characteristics of and the reason for abnormal data, it can be divided into two types: mutated abnormal data and zero-value abnormal data.

Mutated abnormal data. Due to the strong magnetic field interference or human influence on the synchronous monitoring system of transformer DC bias, sudden changes occur in the monitoring data at a certain moment or within a certain period of time, i.e., mutated abnormal data [22]. The mutated abnormal data is divided into three types, as shown in Figure 2, Figure 3 and Figure 4. Figure 2 shows the abnormal noise under the operation of the metro, and mutated abnormal data can occur at the peak and trough of the monitoring data. Figure 3 shows the abnormal noise during the operation of the metro, and the mutated abnormal data occurs under the condition that the DC blocking device is put into operation. Figure 4 shows abnormal data of the neutral DC between the metro shutdown and the operation of the metro, and mutated abnormal data occurs during the metro shutdown.

Based on data for the synchronous monitoring of transformer DC bias in a certain power grid substation from 2019 to 2022, the mutated abnormal data analysis is analyzed as follows:Mutated abnormal data can occur at any working condition and at any time.Mutated abnormal data can occur at a certain time. At this time, both sides of the mutated abnormal data are normal data and the waveform is a spike shape. The mutated abnormal data occur in a certain period of time. At this time, the two sides of the mutated abnormal data are normal data, and the waveform is approximately rectangular.The absolute value of the amplitude is more than 10% of the absolute value of the normal monitoring data on both sides. The absolute value of the amplitude increases significantly.The amplitude of the mutated abnormal data can be close to the amplitude of the monitoring data during the operation of the metro.

Zero-value abnormal data. Communication sub-stations or communication access networks are also prone to switch failures and communication fiber-optic cable disconnections, which make a large amount of monitoring data unable to be centralized and forwarded, and can lead to the generation of zero-value abnormal data [23]; the value of the monitoring data is zero, which is called zero-value abnormal data. Figure 5 shows the zero-value abnormal data of the neutral DC under the operation of the metro.

## 3. Identification Index of Abnormal Data

Based on the above analysis for abnormal data for synchronous monitoring of transformer DC bias, in order to effectively identify different types of abnormal data, based on the abnormal data types and characteristics, this paper proposes three kinds of abnormal data identification indexes.

### 3.1. Synchronous Monitoring of Data Gradient

According to the characteristics of the mutated abnormal data, the gradient of the data for synchronous monitoring is defined as an index to identify mutated abnormal data. By calculating the change in the amount of monitoring data in unit time, the distribution result of the gradient of monitoring data is calculated, and the calculation of the gradient is shown in Equation (1).
(1)$$K_{j}=\frac{w_{j}-w_{j-1}}{t}$$
where, *K_j_* is gradient at time *j*. *t* is the time between time *j* and time *j* − 1. *w_j_* is the monitoring data at time *j*. *w_j_*_−1_ is the monitoring data at time *j* − 1.

### 3.2. Synchronous Monitoring of Data Sliding Kurtosis

In view of the mutation data at a certain time in the mutated abnormal data, the sliding kurtosis is defined to identify such abnormal data [24,25]. In order to make the kurtosis more accurately reflect the changes in a group of data, this paper adopts the method of multi-group sliding calculation. Set the sliding set as *P_i_* = {*x_i_*_−1_,*x_i_*,*x_i_*_+1_}, *i* = 2, 3 … *n*, *P_i_* is the sliding set at position *i*. After the sliding grouping is completed, the kurtosis of the sliding set is calculated one by one. The calculation formula of the sliding kurtosis is shown in Equation (2).
(2)$$Q=\frac{E{\left(x-\mu \right)}^{4}}{\sigma^{4}}$$
where *Q* is the sliding kurtosis, and *x* is the neutral DC, noise and vibration. *μ* is the mean value of *x*, and *σ* is the mean square error of *x*.

### 3.3. Synchronous Monitoring of Data Correlation

A large amount of research has shown that there is a high correlation between neutral DC, vibration and noise under the condition of DC bias [26,27]. For evaluating the Pearson correlation coefficient, the criteria are shown in Table 1 [28].

Therefore, in view of the situation of mutated abnormal data in a certain period of time, the correlation of data for synchronous monitoring is defined as an index, and the Pearson correlation coefficient between the monitoring data is calculated to determine whether there is abnormality in the monitoring data. The calculation formula of the Pearson correlation coefficient is shown in Equation (3).
(3)$$P=\frac{\sum_{i=1}^{n}\left(x_{i}-x_{p}\right)(y_{i}-y_{p})}{\sqrt{\sum_{i=1}^{n}{\left(x_{i}-x_{p}\right)}^{2}}\sqrt{{\sum_{i=1}^{n}\left(y_{i}-y_{p}\right)}^{2}}}$$
where *P* is the Pearson correlation coefficient between monitoring data *x* and monitoring data *y*; *i* takes the value from {1, 2, 3, …, *n*}; *x_i_*, *y_i_* are neutral DC, noise and vibration; *x_p_* is the average value of *x_i_*. *y_p_* is the average value of *y_i_*; the larger the absolute value of *P*, the higher the correlation between *x* and *y*, and the smaller the absolute value of *P*, the lower the correlation between *x* and *y*.

## 4. Abnormal Data Identification Method Based on Multiple Criteria Fusion

### 4.1. Selection Method for the Abnormal Data Identification Criterion Threshold

We collected a large number of synchronous monitoring data of transformer DC bias, and calculated the gradient and sliding kurtosis of the monitoring data. It was found that the change rate of the neutral DC was approximately normal distribution. The sliding kurtosis of the monitoring data generally followed an exponential distribution. Data for the synchronous monitoring of transformer DC bias in a substation was selected for a day; the gradient was calculated using (1) and the sliding kurtosis was calculated using (2), and the distribution of the gradient and sliding kurtosis is shown in Figure 6.

The figure shows that the distribution of the sliding kurtosis of the data for synchronous monitoring is concentrated, so the distribution boundary of the sliding kurtosis is taken as the sliding kurtosis threshold. Pauta criterion can eliminate coarse error data in the sample data subject to normal distribution or approximately normal distribution. This method is simple, and widely used in the field of abnormal data identification [29,30]. Therefore, in this paper, the Pauta criterion is used to select the gradient threshold data for synchronous monitoring of DC bias. The specific calculation method is shown in Equation (4).
(4)$$M=\frac{1}{n}\sum_{i=1}^{n}a_{i}+3\sqrt{\sum_{i=1}^{n}\frac{{\left(a_{p}-\bar{a}\right)}^{2}}{n-1}}$$
where *M* is the threshold for the gradient; *i* takes values from {1, 2, 3, …, *n*}; *a_i_* is the gradient; *a_p_* is the average of *a_i_*.

### 4.2. Abnormal Data Identification Process

The process of the abnormal data identification method includes data collection, using the gradient to identify abnormal data, normal data and suspected abnormal data. Determine the scope of suspected abnormal data and identify suspected abnormal data. Figure 7 shows the flow chart of the abnormal data identification method. Taking suspected abnormal data in noise as an example, the specific steps of the method in this paper are described as follows:

Step1: Collect data for the synchronous monitoring of transformer DC bias, including neutral DC *X_j_*, noise *Y_j_* and vibration *Z_j_*.

Step2: Use the gradient to identify abnormal data, normal data and suspected abnormal data. Calculate the gradient *K_j_*_1_, *K_j_*_2_ and *K_j_*_3_ of neutral DC, noise and vibration. If the gradient is equal to zero, it is judged as zero-value abnormal data. If the gradient is within the threshold range of the gradient and is not zero, it is judged as normal data; otherwise, enter Step3.

Step3: Determine the scope of the suspected abnormal data. If the gradient is not within the threshold range of the gradient, the data is judged as suspected abnormal data. Continue to collect neutral DC, vibration and noise, calculate the gradient, until the gradient between noise *Y_m_* and noise *Y*_(*j*−1)_ is within the threshold range of the gradient; then the noise between *Y_j_* and *Y*_(*m*−1)_ is suspected abnormal data.

Step4: Identify the suspected abnormal data. The suspected abnormal data is a sudden change at a certain moment, and if *m* − 1 = *j*, the sliding kurtosis of the suspected abnormal data with respect to the left and right points is calculated. If the sliding kurtosis is greater than the sliding kurtosis threshold, the data is judged as abnormal data, and, vice versa, the data is judged as normal data. The suspected abnormal data is a sudden change in a certain time period; if *m* − 1 > *j*, the Pearson correlation coefficient between the suspected abnormal data and the vibration and neutral DC is calculated; if the Pearson correlation coefficient is less than 0.2, it is judged as abnormal data, and, vice versa, it is judged as normal data.

## 5. Application Analysis

Taking the abnormal data set for the synchronous monitoring of transformer DC bias in a specific power grid as the object of analysis, the proposed abnormal data identification method was used to identify the different types of abnormal data.

### 5.1. Determination of Threshold of Abnormal Data Identification Index

Based on a large number of normal data for the synchronous monitoring of transformer DC bias, calculating the gradient threshold and sliding kurtosis threshold for different sampling-time data samples, statistics found that the threshold tends to be stable when the sampling time is greater than 3 days, as shown in Figure 8, Figure 9 and Figure 10. Therefore, three days of data for the synchronous monitoring of transformer DC bias are collected in this paper, and the gradient thresholds of neutral DC, vibration and noise are calculated by (4), as shown in Table 2. Step (2) is used to calculate the sliding kurtosis of the all-day data for synchronous monitoring, and the selected threshold of sliding kurtosis is shown in Table 2.

### 5.2. Identification of Mutated Abnormal Data

In this section, the measured mutated abnormal data of a network substation are used for analysis. The noise in a substation from 09:00 to 09:30 on 17 November 2022 is selected. In this period, due to the operation of the high-power frequency conversion equipment, electromagnetic interference is caused to the sensor, and a large amount of mutated abnormal data are generated, as shown in Figure 11.

The suspected abnormal data in the above data were identified based on the gradient criteria in Step 2 and Step 3, and the results were obtained as shown in Figure 12. It is statistically found that the misjudgment rate of the abnormal data identified by Step 2 and Step 3 alone is 9.09%, and the miss rate is 0%. This shows that the suspected abnormal data contains all the abnormal data, but some normal data are misjudged.

In Step 4, the sliding kurtosis index is used to identify the mutation point at a certain time in the above suspected abnormal data, and the Pearson correlation coefficient is used to identify the mutation point at a certain time in the above suspected abnormal data, and the identification results are shown in Figure 13. According to statistics, the identification accuracy of abnormal data reaches 100%, and all abnormal data can be accurately identified without missing judgments.

### 5.3. Result Analysis of Zero-Value Abnormal Data

In this section, the measured zero-value abnormal data of a network substation is used for analysis. Select the neutral DC in a substation in the time period of 13:30~14:30 on 17 November 2022, and when testing data for synchronous monitoring of transformer DC bias in a certain power grid, the neutral DC between 14:00~14:10 is zero, due to the failure of the data transmission line of the current sensor, as shown in Figure 14.

The zero-value abnormal data in the above data can be directly identified by using the gradient criterion in Step 2 of the proposed method, and the obtained results are shown in Figure 15. It is found that the recognition accuracy of zero-value abnormal data is 100%.

### 5.4. Performance Comparison

In order to verify the superiority of the proposed method, we collected a large number of transformer DC bias synchronous monitoring data. However, there was a lack of different types of abrupt abnormal data that occurred over a period of time. Therefore, according to the characteristics of abnormal data, we added some simulated abnormal data on the basis of normal data obtained in real time. The “quartile method” and “Pauta criterion identification method” were compared with the proposed method. The identification results of different methods are shown in Table 3.

Analysis of identification results of zero-value abnormal data. Three methods are used to identify the test data, and the identification results are shown in Figure 16, Figure 17 and Figure 18. It can be seen from Figure 16 that the identification accuracy of the method proposed in this paper is 100% for the zero-value abnormal data contained in the neutral DC, while the identification accuracy of the other two methods is 0%. It can be seen from Figure 17 that the identification accuracy of the three methods for the zero-value abnormal data contained in the noise is 100%. It can be seen from Figure 18 that the identification accuracy of the proposed method and “Pauta criterion identification method” for the zero-value abnormal data contained in the vibration is 100%, while the identification accuracy of the “quartile method” is 0%. According to the above results, the neutral DC and vibration contain a large amount of data close to 0. The amplitude distribution of zero-value abnormal data is close to that of normal data, while the normal data of noise is very different from that of the zero-value abnormal data. This will lead to the “quartile method” and the “Pauta criterion identification method” judging the zero-value abnormal data in the neutral DC and vibration as normal data, and the zero-value abnormal data in the noise as abnormal data. However, the proposed method can accurately identify the zero-value abnormal data in neutral DC, vibration and noise.Analysis of identification results of mutated abnormal data. Similarly, three methods are used to identify the test data, and the identification results are shown in Figure 19, Figure 20 and Figure 21. According to the statistical analysis above, the identification results are shown in Table 3. It can be seen from Table 3 that the method proposed in this paper can effectively identify the mutated abnormal data, and the misjudgment rate and missed judgment rate of neutral DC, vibration and noise are significantly lower than that of the “quartile method” and “Pauta criterion identification method”. It can be seen that the method proposed in this paper can accurately identify different types of abnormal data, which is more practical and applicable. According to the above results, the amplitude between some mutated abnormal data and normal data is narrow, which will lead to the omission of some mutated abnormal data by the “quartile method” and the “Pauta criterion identification method”. The synchronous monitoring data of the transformer DC bias approximately obeys normal distribution, so using only the “Pauta criterion identification method” for identification will lead to misjudgment.

### 5.5. Applicability Analysis of Identification Method

#### 5.5.1. Influence of Different Network Monitoring Data on Identification Results

To verify the applicability of the proposed method to the synchronous monitoring data of transformer DC bias in different slices of the network, we select the synchronous monitoring abnormal data of transformer DC bias of power grid A for simulation, which is different from the above-mentioned power grid. The identification results are shown in Figure 22, Figure 23 and Figure 24. From the figures, it can be seen that the proposed method in this paper can accurately identify the abnormal data of this power grid. For the synchronous monitoring abnormal data of transformer DC bias of different grids there are commonly mutated abnormal data and zero-value abnormal data, and their data characteristics are the same as the abnormal data characteristics used in this paper. Therefore, the method proposed in this paper is applicable to the synchronous monitoring data of transformer DC bias of any power grid.

#### 5.5.2. Influence of Increased Abnormalities on Identification Results

Consider the applicability of the proposed method when the abnormalities considered are increased. The method proposed in this paper is real-time judgment, that is, the synchronous monitoring data of the transformer DC bias at each moment are judged in turn. Therefore, when the number of abnormal data increases, the method proposed in this paper is still applicable. At the same time, since the method proposed in this paper is based on the characteristics of zero-value abnormal data and mutated abnormal data, monitoring data containing other types of abnormal data does not conform to the characteristics of zero-value abnormal data and mutated abnormal data. Therefore, when the other types of abnormal data in the monitoring data increase, the method proposed in this paper is still applicable.

## 6. Conclusions

The proposed method is suitable for a synchronous and real-time transformer DC bias monitoring system. In this paper, based on the actual abnormal data characteristics, an abnormal data identification method and abnormal data identification process are proposed. The application results of the method provide the following conclusions:Based on the measured data results in a certain power grid, the verification shows that the accuracy of the proposed method in identifying mutated abnormal data and zero-value abnormal data is claimed to be 100%.Compared with the traditional “quartile method” and “Pauta criterion identification method”, the accuracy of the proposed method is claimed to be significantly higher than that of the above methods, which indicates that the proposed method makes up for the shortcomings of the traditional algorithm, and it is claimed that the proposed method has high practicability and adaptability.

## Figures and Tables

**Figure 1 sensors-23-04959-f001:**
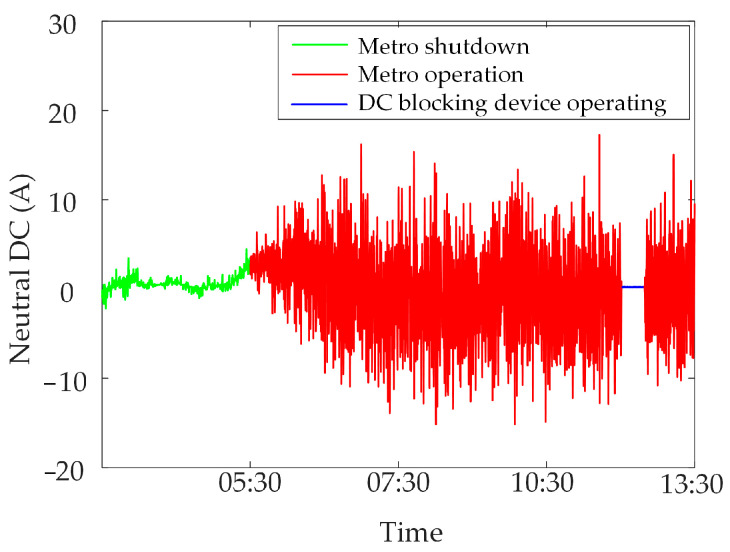
Neutral DC under different operating conditions.

**Figure 2 sensors-23-04959-f002:**
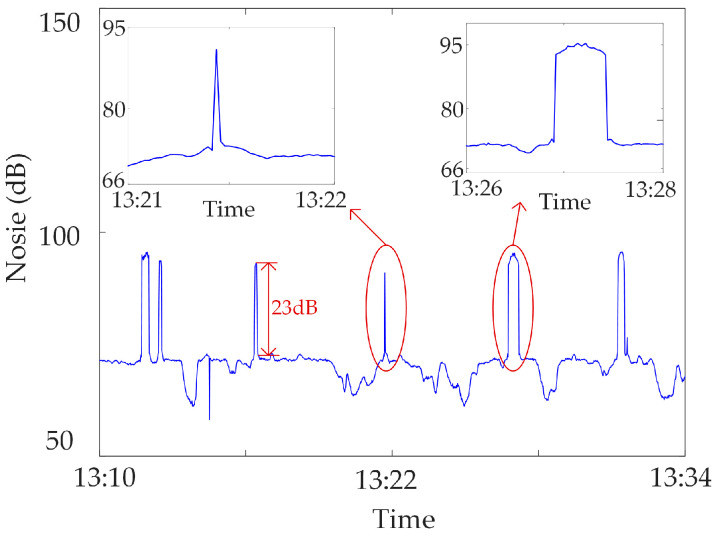
Mutated abnormal data under operating conditions of metro.

**Figure 3 sensors-23-04959-f003:**
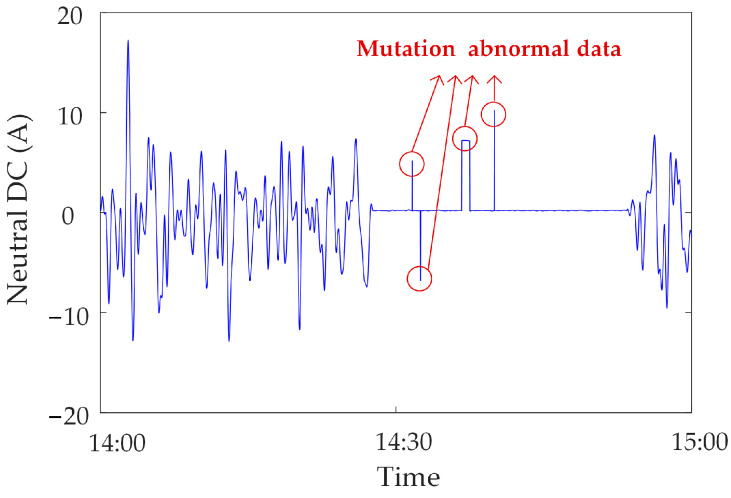
Mutated abnormal data under operating conditions of DC blocking device.

**Figure 4 sensors-23-04959-f004:**
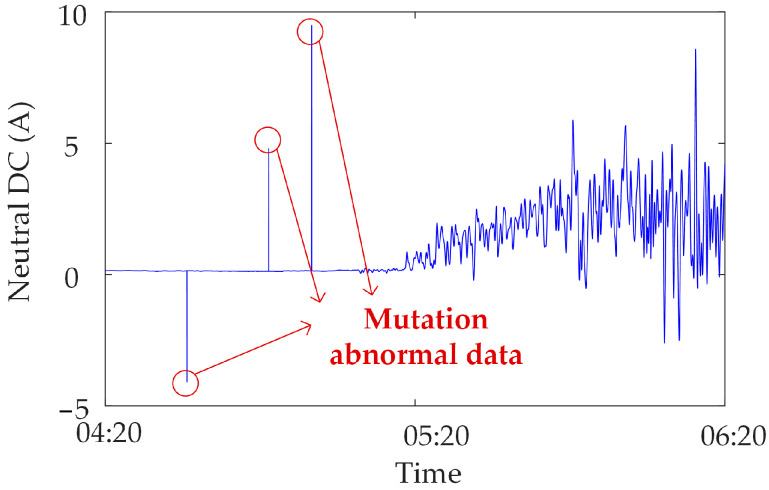
Mutated abnormal data under the condition of metro shutdown.

**Figure 5 sensors-23-04959-f005:**
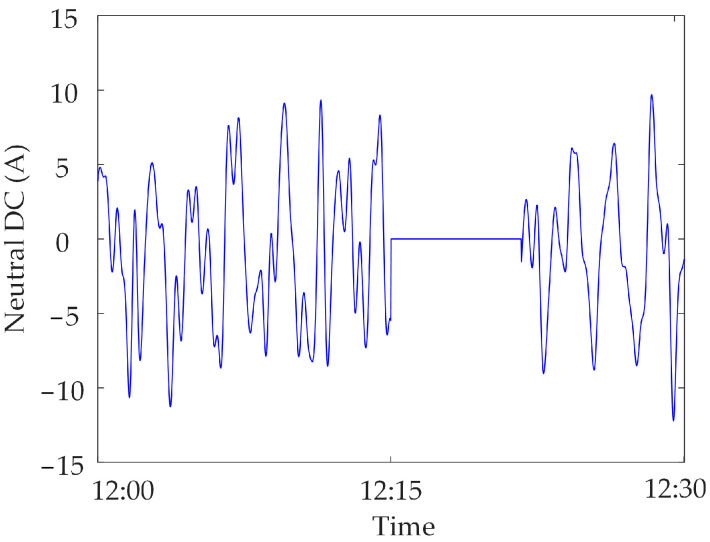
Zero-value abnormal data of neutral DC in a substation.

**Figure 6 sensors-23-04959-f006:**
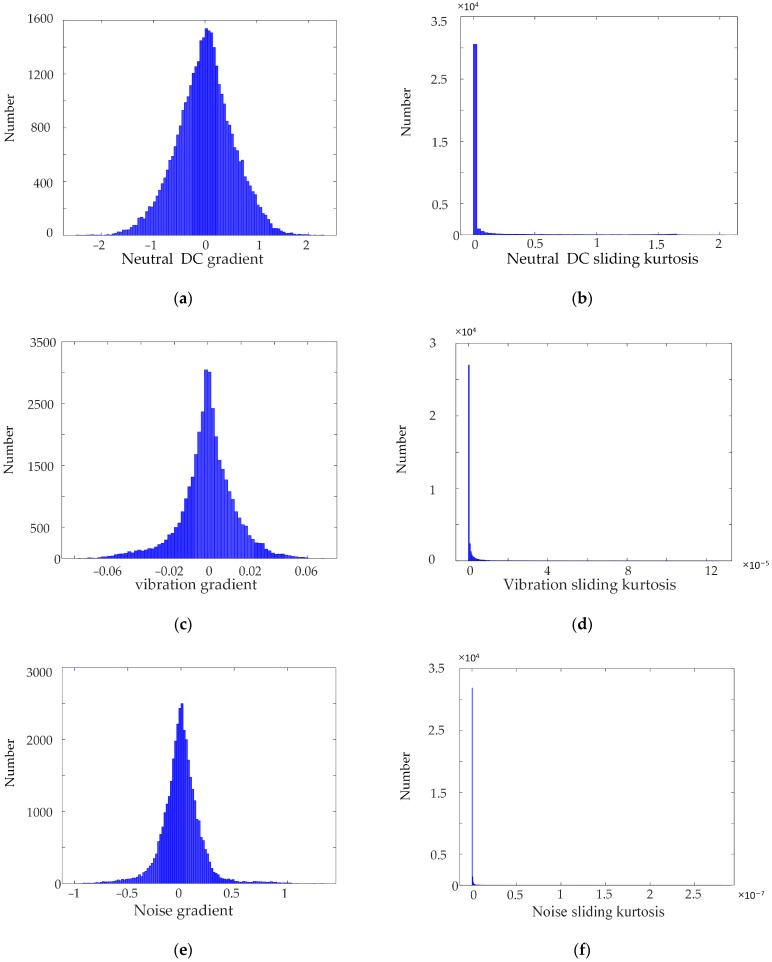
Gradient and sliding kurtosis distribution of data for synchronous monitoring. (**a**) Distribution of neutral DC gradient. (**b**) Distribution of neutral DC sliding kurtosis. (**c**) Distribution of vibration gradient. (**d**) Distribution of vibration sliding kurtosis. (**e**) Distribution of noise gradient. (**f**) Distribution of noise sliding kurtosis.

**Figure 7 sensors-23-04959-f007:**
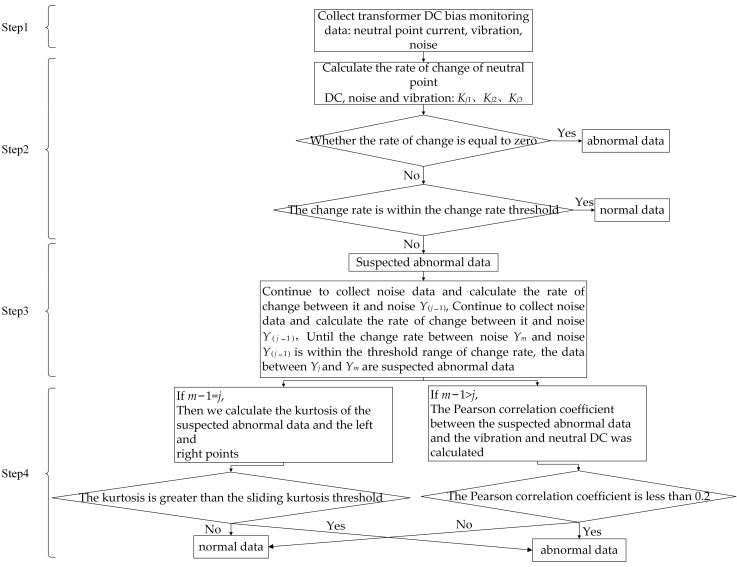
Abnormal data identification flow chart.

**Figure 8 sensors-23-04959-f008:**
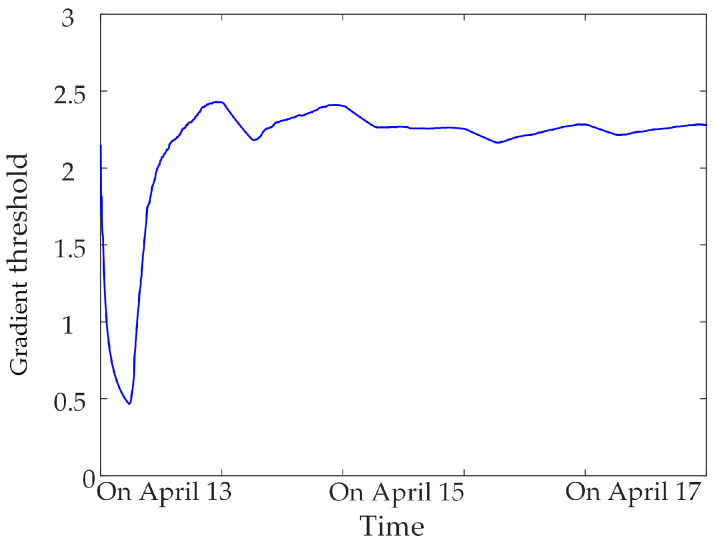
Neutral DC gradient threshold.

**Figure 9 sensors-23-04959-f009:**
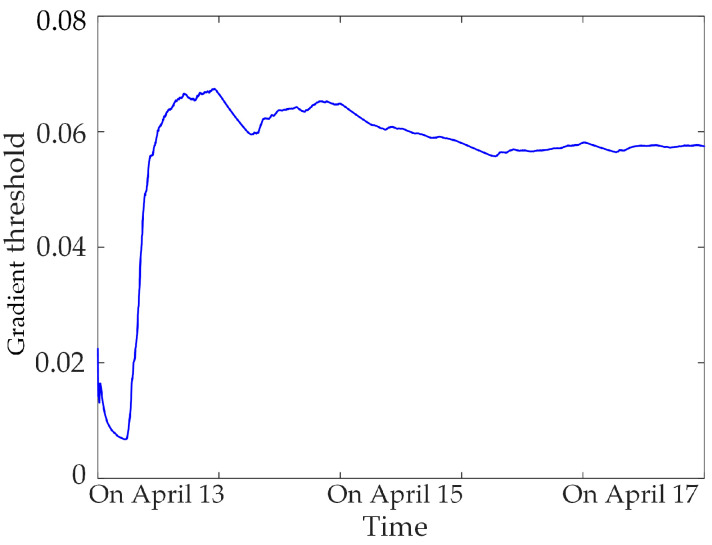
Vibration gradient threshold.

**Figure 10 sensors-23-04959-f010:**
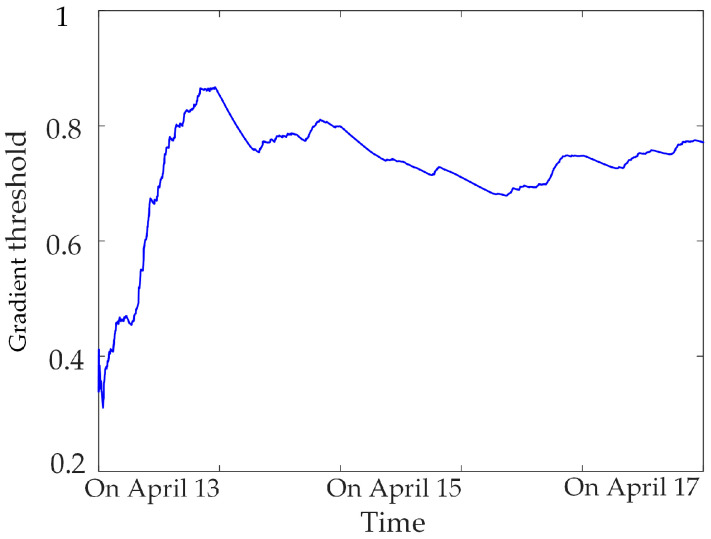
Noise gradient threshold.

**Figure 11 sensors-23-04959-f011:**
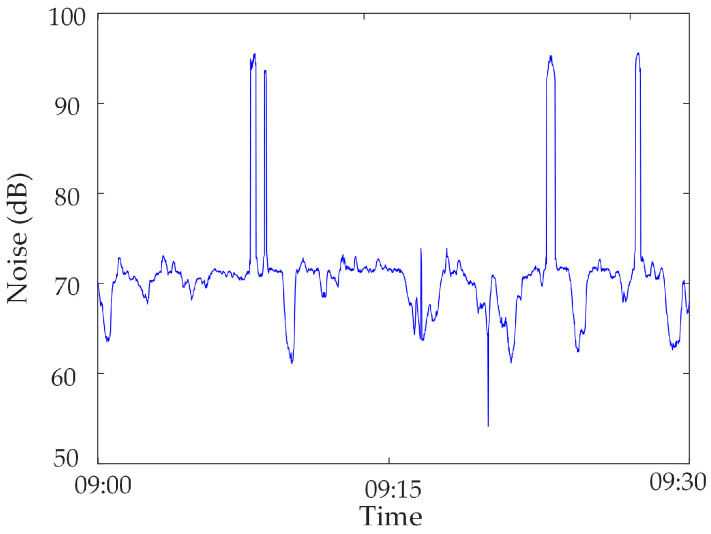
Mutated abnormal data in noise.

**Figure 12 sensors-23-04959-f012:**
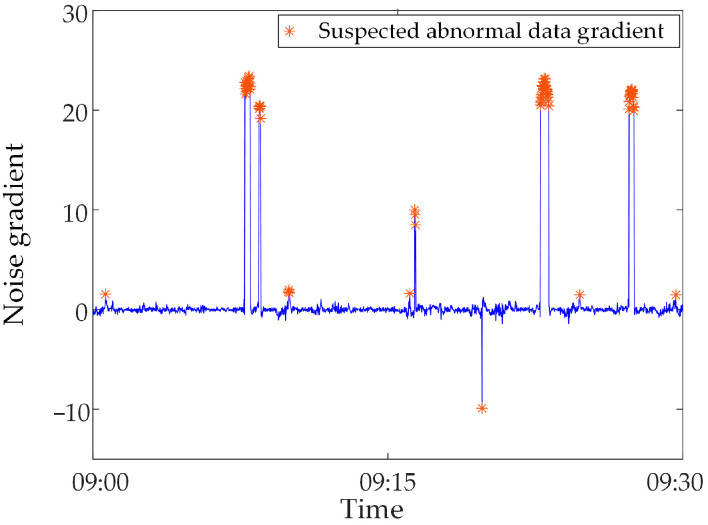
Suspected abnormal data identification results.

**Figure 13 sensors-23-04959-f013:**
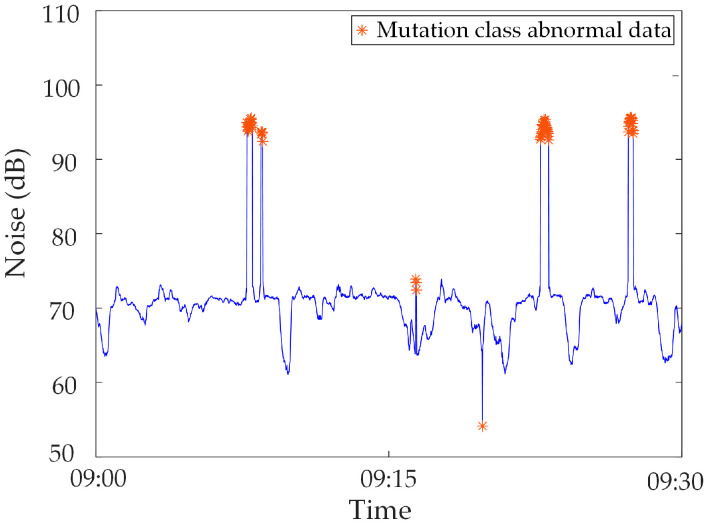
Mutated abnormal data identification results.

**Figure 14 sensors-23-04959-f014:**
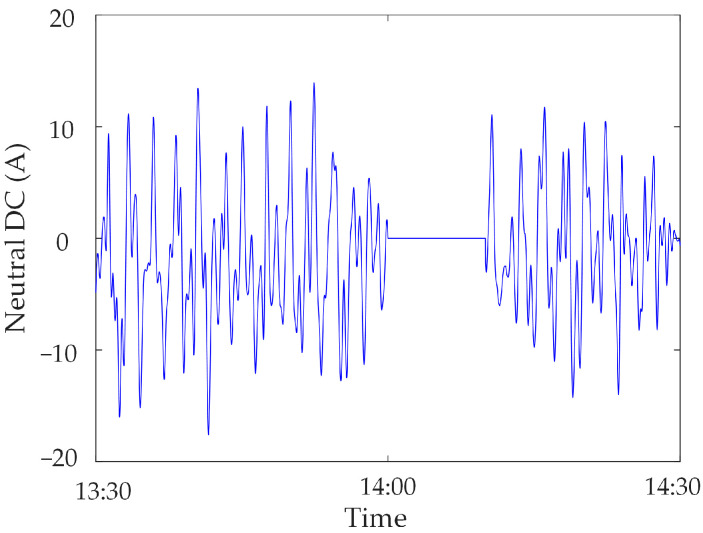
Zero-value abnormal data in Neutral DC.

**Figure 15 sensors-23-04959-f015:**
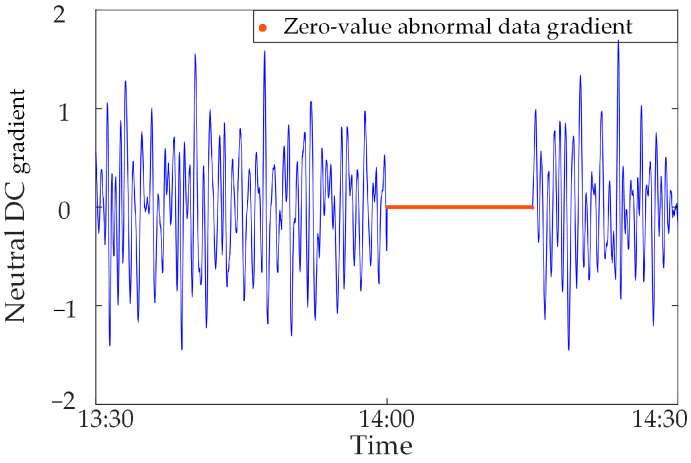
Zero-value abnormal data identification results.

**Figure 16 sensors-23-04959-f016:**
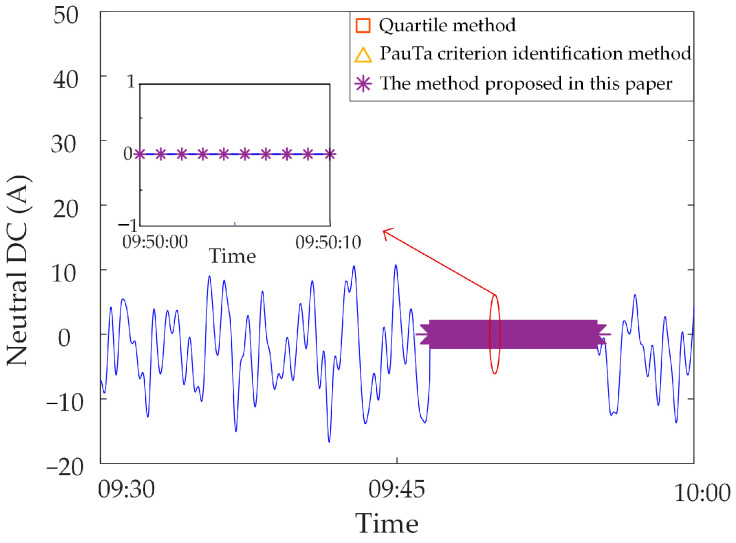
Results of different identification methods for zero-value abnormal data in neutral DC.

**Figure 17 sensors-23-04959-f017:**
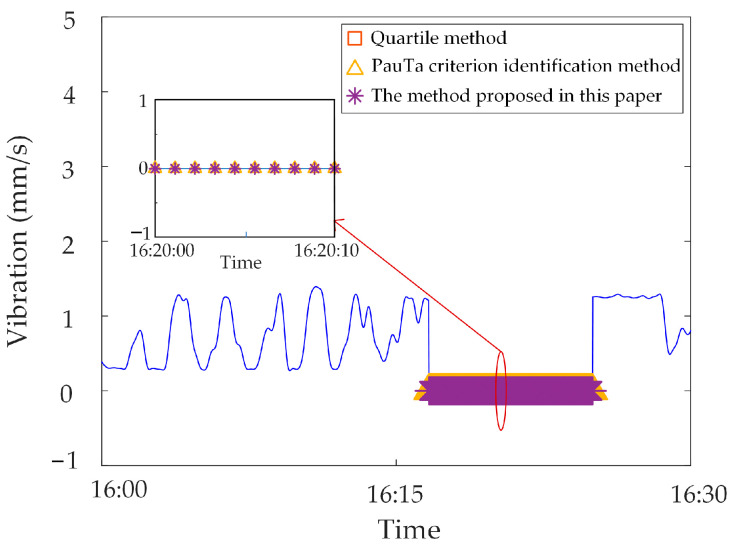
Results of different identification methods for zero-value abnormal data in vibration.

**Figure 18 sensors-23-04959-f018:**
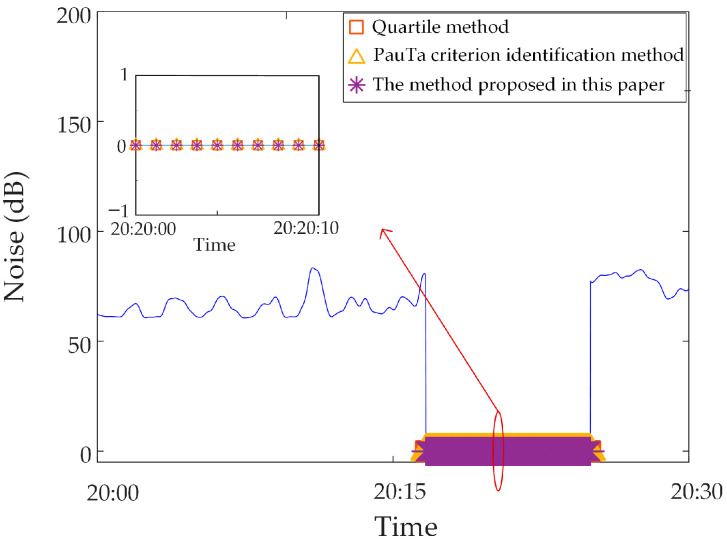
Results of different identification methods for zero-value abnormal data in noise.

**Figure 19 sensors-23-04959-f019:**
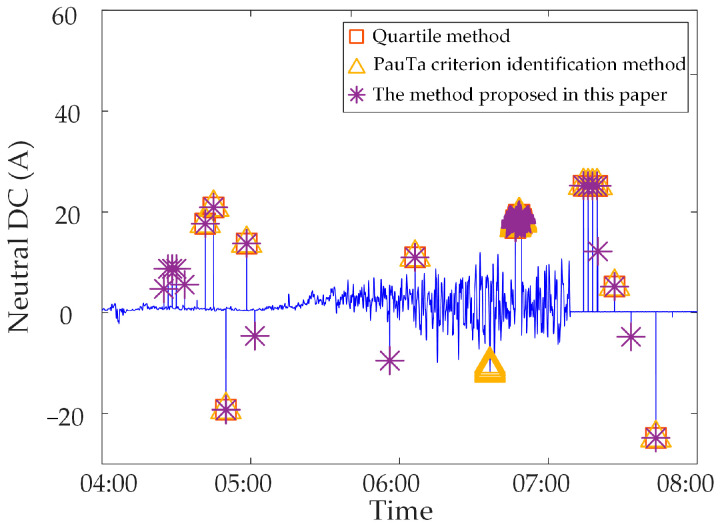
Results of different identification methods for mutated abnormal data in neutral DC.

**Figure 20 sensors-23-04959-f020:**
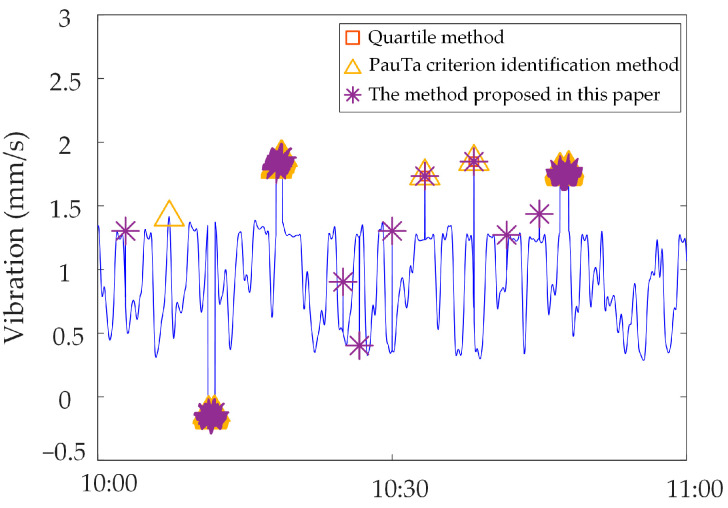
Results of different identification methods for mutated abnormal data in vibration.

**Figure 21 sensors-23-04959-f021:**
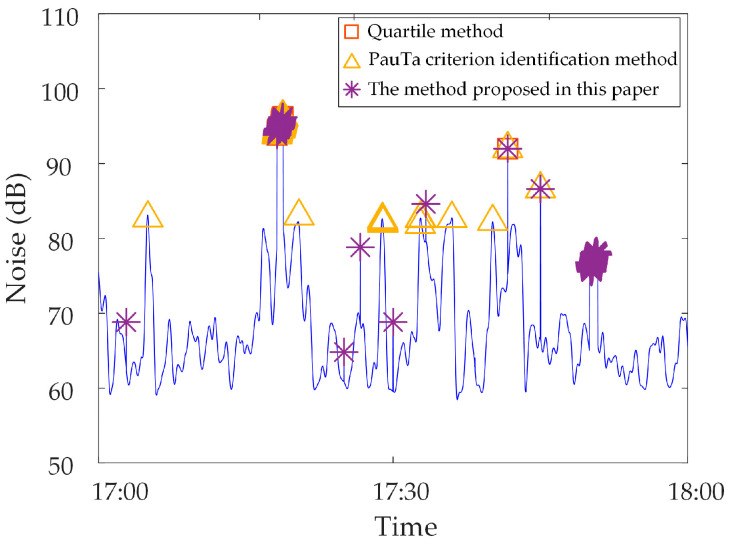
Results of different identification methods for mutated abnormal data in noise.

**Figure 22 sensors-23-04959-f022:**
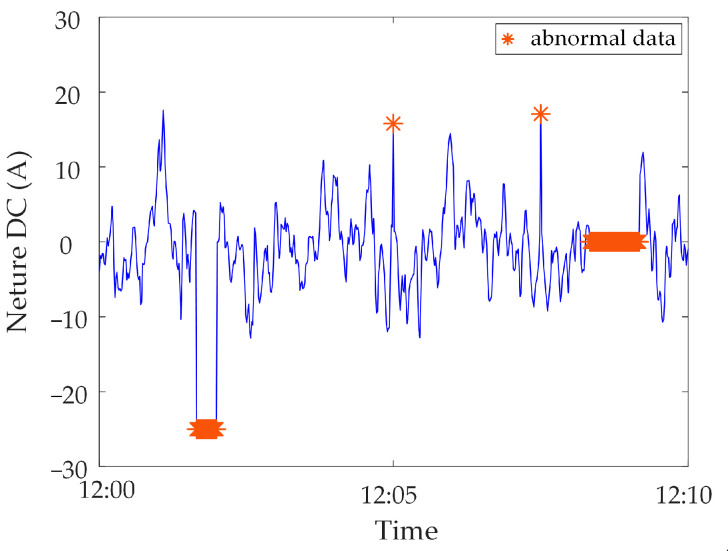
Identification results of abnormal neutral DC data of a substation in the power grid A.

**Figure 23 sensors-23-04959-f023:**
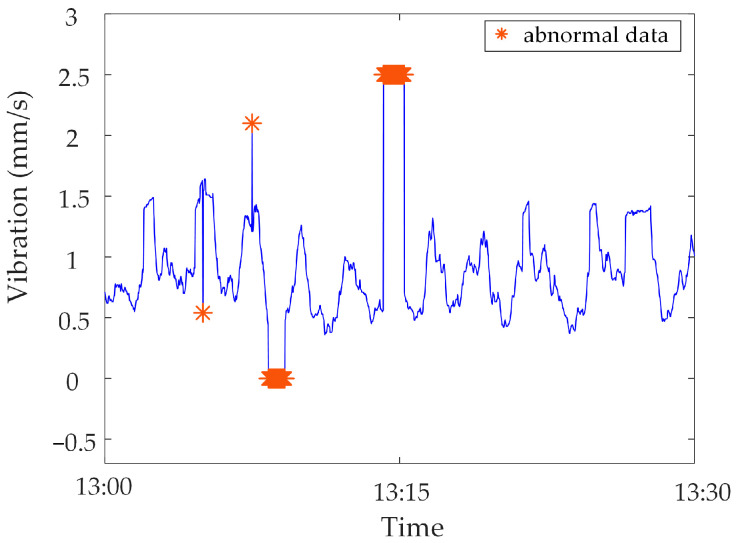
Identification results of abnormal vibration data of a substation in the power grid A.

**Figure 24 sensors-23-04959-f024:**
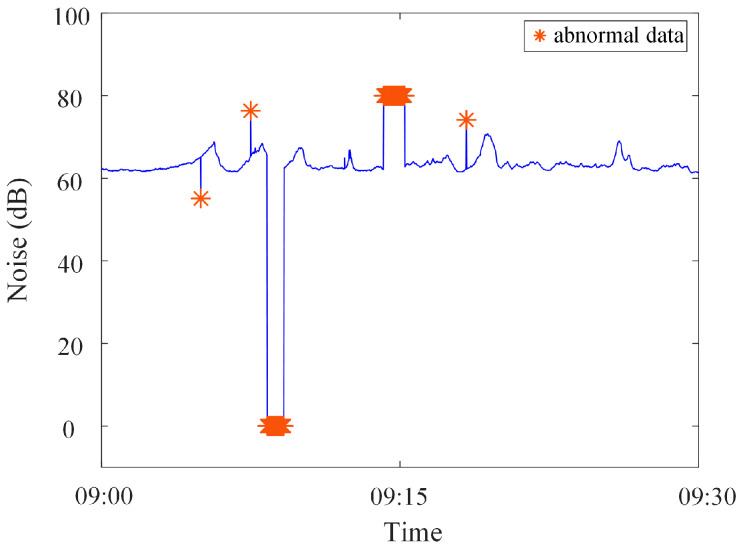
Identification results of abnormal noise data of a substation in the power grid A.

**Table 1 sensors-23-04959-t001:** The degree of association of the Pearson correlation coefficient.

Degree of Correlation	Range of *P*
Very weakly correlated or irrelevant	0.0 ≤ *P* < 0.2
Weak correlation	0.2 ≤ *P* < 0.4
Moderate degree of correlation	0.4 ≤ *P* < 0.6
Strong correlation	0.6 ≤ *P* < 0.8
Highly correlated	0.8 ≤ *P* ≤ 1.0

**Table 2 sensors-23-04959-t002:** Thresholds for gradient and sliding kurtosis.

Monitoring Data	Gradient	Sliding Kurtosis
Neutral DC	±2.5	1
Vibration	±0.07	2 × 10^−5^
Noise	±0.8	1 × 10^−7^

**Table 3 sensors-23-04959-t003:** Comparison of identification results.

Identification Method		Misjudgment Rate/%	Missed Judgment Rate/%
Neutral DC	Proposed method	0.0	0.0
	Quartile method	0.0	6.4
	Pauta criterion identification method	2.3	6.4
Vibration	Proposed method	0.0	0.0
	Quartile method	0.0	3.5
	Pauta criterion identification method	0.7	3.5
Noise	Proposed method	0.0	0.0
	Quartile method	0.0	59.4
	Pauta criterion identification method	4.2	45.8

## Data Availability

The datasets generated for this study are available on request to the corresponding author.

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
