# Peer review of "Identification of Abnormal Data for Synchronous Monitoring of Transformer DC Bias Based on Multiple Criteria"

_sensors, 2023, doi:10.3390/s23104959_

Round 1
Reviewer 1 Report
In this paper, based on the actual abnormal data characteristics, an abnormal data identification menthod and abnormal data identification process are proposed. The application results of the method provide the following conclusions:
1. Based on the measured data result in a certain power grid, the verification shows that the accuracy of the proposed method in identifying mutation abnormal data and zero-value abnormal data is claimed to be 100%.
2. Compared with the traditional "Quartile method" and "PauTa criterion identification method", the accuracy of the proposed method is claimed to be significantly higher than that of the above methods, which indicates that the proposed method makes up for the shortcomings of the traditional algorithm, and it is claimed that the proposed method has high practicability and adaptability.
As the data pertains to a certain power grid, whether the proposed method can be applicable to any other power grid? If not, what are the changes to be brought to have a generalised method.
Author Response
Point 1: In this paper, based on the actual abnormal data characteristics, an abnormal data identification method and abnormal data identification process are proposed. The application results of the method provide the following conclusions:
- Based on the measured data result in a certain power grid, the verification shows that the accuracy of the proposed method in identifying mutation abnormal data and zero-value abnormal data is claimed to be 100%.
- Compared with the traditional "Quartile method" and "PauTa criterion identification method", the accuracy of the proposed method is claimed to be significantly higher than that of the above methods, which indicates that the proposed method makes up for the shortcomings of the traditional algorithm, and it is claimed that the proposed method has high practicability and adaptability.
As the data pertains to a certain power grid, whether the proposed method can be applicable to any other power grid? If not, what are the changes to be brought to have a generalised method.
Response 1: We are very grateful to the Editor for reading carefully. According to the Editor ’s review, we have revised the expression of the conclusion in Section 6 of the revised manuscript. The method proposed in this paper is applicable to any other power grid. For the synchronous monitoring abnormal data of transformer DC bias in different power grids, there are common mutation abnormal data and zero abnormal data, and the data characteristics are the same as those used in this paper. We collected a large number of synchronous monitoring data of transformer DC bias from Guiyang, Chengdu, Shenzhen and other cities in China. We simulate and analyze the synchronous monitoring data of transformer DC bias, and the results show that the proposed method can accurately identify abnormal data. The abnormal data identification results of a substation in Shenzhen and Chengdu, China are shown in Figure 1-Figure 6. It can be seen that the method proposed in this paper is suitable for synchronous monitoring data of transformer DC bias in any power grid.
According to the Editor ’s review, we have added Section 5.5.1 in the revised manuscript to supplement the application of the method proposed to other power grids.
Figure 1. Identification results of noise abnormal data in a substation in Shenzhen, China.
Figure 2. Identification results of neutral DC abnormal data in a substation in Shenzhen, China.
Figure 3. Identification results of abnormal vibration data of a substation in Shenzhen, China.
Figure 4. Identification results of neutral DC abnormal data in a substation in Chengdu, China.
Figure 5. Identification results of abnormal vibration data of a substation in Chengdu, China.
Figure 6. Identification results of noise abnormal data in a substation in Chengdu, China.

Reviewer 2 Report
comments:
However, the topic is interesting "Identification of Abnormal Data for SynchronousMonitoringof Transformer DC Bias Based on Multiple Criteria" there are some comments and issues that need to be clarified
1. In the abstract, the authors need to accurately define the goal of the research such as this paper is to...
2. In the abstract, "The results show that the proposed method can accurately and reliably identify different types of abnormal data. Findings can be more specific
3. In the introduction, the literature review can include more arguments required.
4. Results, Authors can include a discussion section that presents a summary discussion of the results and point out findings.
5. References, please look for diversity in the authors who investigated the topic.
Author Response
Point 1: In the abstract, the authors need to accurately define the goal of the research such as this paper is to...
Response 1: We are very grateful to the Editor for their careful reading and valuable comments. According to the Editor ’s review, we have defined the research objectives accurately in the abstract of the revised manuscript. The revised part has been marked red in the revised manuscript.
Point 2 In the abstract, "The results show that the proposed method can accurately and reliably identify different types of abnormal data. Findings can be more specific
Response 2: We are very grateful to the Editor for their careful reading and valuable comments. According to the Editor ’s review, we have made specific additions to the findings in the abstract of the revised version. The revised part has been marked red in the revised manuscript.
Point 3: in the introduction, the literature review can include more arguments required.
Response 3: We are very grateful to the Editor for their careful reading and valuable comments. According to the Editor's review, we have added references as arguments in paragraph 2 Section 1 of the revised manuscript. The revised part has been marked red in the revised manuscript.
Point 4: Results, Authors can include a discussion section that presents a summary discussion of the results and point out findings.
Response 4: We are very grateful to the Editor for their careful reading and valuable comments. According to the Editor's review, we have summarized and discussed the results and pointed out the findings in paragraph 2 and3 Section 5.4 of the revised manuscript. The revised part has been marked red in the revised manuscript.
Point 5: References, please look for diversity in the authors who investigated the topic.
Response 5: We are very grateful to the Editor for their careful reading and valuable comments. According to the Editor's review, we have added references in paragraph 1 Section 1, paragraph 1 Section 3.2 and paragraph 2 Section 4.1 of the revised manuscript. The revised part has been marked red in the revised manuscript.

Reviewer 3 Report
Why neutral dc gradient data is normally distributed
The paper is well-written and must be accepted.
Author Response
Point 1: Why neutral dc gradient data is normally distributed?
Response 1: We are very grateful to the Editor for reading carefully. We collected a large number of synchronous monitoring data of transformer DC bias, and calculated the change rate of monitoring data. It is found that the change rate of neutral DC is approximately normal distribution. Because the metro operation will lead to the increase of the neutral DC of the transformer, which will lead to the DC bias of the transformer. The traction current of the train is shown in Fig.1.During the idle period of the train, the traction current is 200 A, which is only powered by the electric equipment of the train lighting. When the traction current is greater than 200 A, the train is in the traction process. When the traction current is less than 200A, the train is in the braking process. It can be seen from the figure that the proportion of traction and braking during train operation is the same. This will lead to the same trend of increase and decrease of transformer neutral DC, so the change rate of neutral DC will be symmetrically distributed and approximately normal distribution.
According to the Editor's review, we have revised the expression in paragraph 1 Section 4.1 of the revised manuscript. The revised part has been marked red in the revised manuscript.
Figure 1. Train running time-traction current curve.

Reviewer 4 Report
In this paper to ensure the reliability and effectiveness of synchronous monitoring data, an abnormal data identification method is proposed. By analyzing the abnormal data of different types, the characteristics of abnormal data are obtained. Based on this, the abnormal data identification indexes are introduced, including gradient, sliding kurtosis and Pearson correlation coefficient. Firstly, the PauTa criterion is used to determine the threshold of the gradient index. Then, gradient is used to identify the suspected abnormal data. Finally, the sliding kurtosis and Pearson correlation coefficient are used to identify the abnormal data. Data for synchronous monitoring of transformer DC bias in a certain power grid is used to verify the proposed method. The authors claim that the proposed method can accurately and reliably identify different types of abnormal data.
The authors shall clarify the following :
Whether the abnormal data are obtained from simulation or real time?
whether the proposed method may be applicable for any kind of practical system?
While applying the proposed system any assumptions made?
If the abnormalities considered is increased whether the proposed method will be suitable?
Minor spelling check/English corrections are needed. For example, in abstract.., spelling for ' method ' is wrongly given..
Therefore, to ensure the reliablility and effectiveness of synchronous
monitoring data, an abnormal data identification menthod is proposed.
After doing revision it shall be considered for publication.
Author Response
Point 1: Whether the abnormal data are obtained from simulation or real time?
Response 1: We are very grateful to the Editor for reading carefully. In Section 5.2 and Section 5.3, this paper uses real-time DC bias synchronous monitoring data. In Section 5.4, due to the lack of different types of mutational anomaly data that concentrated in a period of time., we add some simulated abnormal data based on the characteristics of abnormal data based on the normal data obtained in real time. Finally, the simulation results show that the proposed method can accurately identify real-time data and simulated abnormal data.
According to the Editor's review, we have supplemented the source of abnormal data in paragraph 1 Section 5.2, paragraph 1 Section 5.3 and paragraph 1 Section 5.4 of the revised manuscript. The revised part has been marked red in the revised manuscript.
Point 2: whether the proposed method may be applicable for any kind of practical system?
Response 2: We are very grateful to the Editor for reading carefully. The method proposed in this paper is not applicable to any practical system. The abnormal data identification method proposed in this paper is aimed at the real-time acquisition of transformer DC bias synchronous monitoring data. Therefore, the proposed method is suitable for a synchronous and real-time transformer DC bias monitoring system.
According to the Editor's review, we made a supplementary explanation on the system applicable to the method proposed in paragraph 1 Section 1 and in paragraph 1 Section 6 of the revised manuscript. The revised part has been marked red in the revised manuscript.
Point 3: While applying the proposed system any assumptions made?
Response 3: We are very grateful to the Editor for reading carefully. The abnormal data identification method proposed in this paper is aimed at the real-time acquisition of transformer DC bias synchronous monitoring data. Therefore, we apply the proposed system assumption to a synchronous and real-time transformer DC bias monitoring system.
According to the Reviewer's questions, We have provided supplementary explanations of the assumptions made in the application of the proposed system in paragraph 1 Section 1 and in paragraph 1 Section 6 of the revised manuscript. The revised part has been marked red in the revised manuscript.
Point 4: If the abnormalities considered is increased whether the proposed method will be suitable?
Response 4: We are very grateful to the Editor for reading carefully. When the abnormal data increases, the method proposed in this paper is still applicable. The method proposed in this paper is real-time judgment, that is, the synchronous monitoring data of transformer DC bias at each moment are judged in turn. Therefore, when the number of abnormal data increases, the method proposed in this paper is still applicable. At the same time, because the method proposed in this paper is based on the characteristics of zero-value abnormal data and mutation abnormal data, the monitoring data containing other types of abnormal data does not conform to the characteristics of zero-value abnormal data and mutation abnormal data. Therefore, when the types of other types of abnormal data in the monitoring data increase, the method proposed in this paper is still applicable.
According to the Editor ’s review, we have added Section 5.5.2 to the revised manuscript, explaining that the method proposed is still applicable when the abnormalities considered is increased. The revised part has been marked red in the revised manuscript.
Point 5: Minor spelling check/English corrections are needed. For example, in abstract.., spelling for ' method ' is wrongly given..
Response 5: We are very grateful to the Editor for their careful reading and valuable comments. According to the Editor's review, we have carefully examined the English spelling problems and corrected them. The modified content has been marked red in the manuscript.
